# Development and Validation of the Values Internalization Scale

**DOI:** 10.3390/bs15050660

**Published:** 2025-05-12

**Authors:** Lanting Wu, Youguo Chen, Xiting Huang

**Affiliations:** 1Faculty of Psychology, Southwest University, Chongqing 400715, China; wlanting@163.com; 2Research Center for Psychology and Social Development, Southwest University, Chongqing 400715, China

**Keywords:** values, internalization, scale development, reliability, validity

## Abstract

Clarifying the stages of the values internalization process enables a more effective selection of interventions to promote targeted internalization. However, no specialized instrument currently exists to measure this process. Grounded in the four-stage model of values internalization, this study aimed to develop and validate a Values Internalization Scale (VIS) for adults. Data from Sample 1 (*N* = 474) were subjected to item analysis and exploratory factor analysis, yielding a scale with a four-factor structure. Subsequently, confirmatory factor analysis and reliability and validity tests were conducted using Sample 2 (*N* = 470). The results indicated that the four-factor model demonstrated a good model fit and that the scale exhibited satisfactory internal consistency reliability, criterion-related validity, and test–retest reliability. The final 25-item VIS comprises four dimensions: ignoring-resistance, understanding, attempt to practice, and integration stages. The 25-item VIS developed in this study performed exceptionally well regarding reliability and validity and can be utilized in subsequent research on the internalization of values.

## 1. Introduction

Values are standards that individuals use to select and interpret behaviors and evaluate people (including oneself) and events ([56]). Values not only guide decisions in areas such as career, education, and romantic relationships but also shape an individual’s behaviors, habits, and personality traits ([39]). Schwartz’s value theory ([56]), currently the most well-regarded evidence-based theory on values, identifies ten fundamental human values: self-direction, stimulation, hedonism, achievement, power, security, tradition, conformity, benevolence, and universalism ([56]). Cross-cultural research has shown that Schwartz’s values are universally recognized and have near-universal meaning across culturally diverse groups ([17]; [57]). Many studies on value change have adopted this theory as their theoretical framework ([5]; [17]). The Schwartz value survey (SVS), developed based on Schwartz’s value theory, is one of the most widely used instruments to assess personal values.

Scholars often use the term “internalization” to describe the process by which individuals adopt societal values ([18]). As a core mechanism of socialization ([19]), internalization of values (values internalization) is defined as “taking over the values and attitudes of society as one’s own so that socially acceptable behavior is motivated not by anticipation of external consequences but by intrinsic or internal factors” ([26]). Values internalization is relevant across various domains. In education, internalizing learning-related values is a crucial process for student development and academic success, and fostering values internalization can enhance students’ academic performance, well-being, and engagement ([71]). In cultural research, the degree of internalization of cultural beliefs is closely linked to an individual’s psychological well-being ([42]). Such evidence shows that investigating values internalization not only provides deeper insights into the psychological mechanisms underlying values formation but also offers theoretical and practical implications for promoting personal growth, social adaptation, and mental health. Despite its theoretical significance, research on values internalization has largely concentrated on its outcomes or mechanisms, with limited attention paid to the distinct stages that individuals experience as they internalize values ([40]; [2]). Furthermore, there is a notable lack of psychometrically robust instruments for assessing individual positions within this developmental trajectory. Existing values-related measures, such as the SVS ([56]) or portrait values questionnaire ([61]), primarily assess values content or priorities but do not capture the dynamic and progressive nature of internalization. To address this gap, this study aims to develop and validate the Values Internalization Scale (VIS).

### 1.1. Theories About the Process of Values Internalization

A review of the relevant literature was conducted before applying methodological procedures and statistical analyses to identify an appropriate theoretical framework for the values internalization process. This step provided a solid conceptual foundation for the subsequent identification of potential dimensions and the development of initial items ([10]). Several theoretical perspectives have delineated stage-based models to characterize the internalization process from an observer’s perspective. Vygotsky’s sociocultural theory, which explores cognitive development in children, proposes that psychological functional development is achieved through internalization. [14] ([14]) summarized Vygotsky’s staged model of internalization, defining it as a sequential process comprising four stages: (1) a “primitive” level, (2) a “naïve” level, (3) external cultural mediation, and (4) internalization of the mediational link. [33] ([33]) proposed a three-stage model of social influence through studies of extreme ideological shifts (e.g., religious conversions) from a motivational perspective: (1) compliance, (2) identification, and (3) internalization. Similarly, [20] ([20]) developed the self-determination theory (SDT) based on their laboratory research; SDT distinguishes between introjection (individuals who conform to values and rules under external motivation) and integration (individuals who assimilate values and rules under internal motivation and act autonomously). Subsequent extensions of SDT have further categorized external motivation internalization into four phases: (1) lack of internalization, (2) partial internalization, (3) identified internalization, and (4) integrated internalization ([71]; [72]). 

These internalization theories provide insights into the internalization of values. However, these studies have not directly explored the process of values internalization or classified it from an external perspective. In individual cognition, it is essential not only to interpret others’ behaviors from an observer’s standpoint but also to analyze one’s actions from the actor’s perspective. The same event may be interpreted in strikingly different ways, depending on these two perspectives ([41]). Furthermore, subjective conscious experiences require a first-person approach to capture the nuances of lived experiences ([38]). A qualitative study by [75] ([75]) conducted a groundbreaking qualitative inquiry into values internalization using an actor-centered approach. Using in-depth interviews with a heterogeneous sample of adults across ages and professions, they reconstructed four sequential phases: ignoring-resistance, understanding, attempt to practice, and integration stages (see Figure 1). These stages exhibit distinct characteristics across the cognitive, behavioral, motivational, emotional, and self-related dimensions (See Table 1 for details). In the ignoring-resistance stage, individuals either remain unaware of the significance of certain values or actively resist adopting them. Notably, this stage has been largely overlooked in previous internalization theories, which tend to focus on either the initiation of internalization or low levels of internalization (e.g., [33] ([33]) theory of compliance, or the lack-of-internalization stage in [20] ([20]) SDT). When values-related information is neglected, it fails to enter an individual’s information processing system ([11]). Moreover, when individuals perceive resistance to external information as justified, they are more likely to reject persuasive attempts by external sources ([7]; [54]; [74]). Consequently, this unique model reveals that overcoming neglect resistance is the gateway to internalizing values and must penetrate attentional filters before subsequent assimilation. By contrast, observer-oriented theories overlook this critical threshold process. 

### 1.2. Measurements of Internalization Values

In addition to the theoretical frameworks of the values internalization stages, scholars have conducted empirical measurements of values internalization. Existing studies have employed behavioral experiments and questionnaire-based assessments, with questionnaires being the most widely used instrument. The questionnaires can be broadly categorized into two types. The first category derives measurement indicators from the definition of values internalization by assessing them through motivation ([44]; [50]; [49]; [51]; [53]), behavior ([69]), and cognition ([32]). The development and application of these measures have enriched and expanded research on values internalization. However, this study had several limitations. First, using motivation, behavior, and cognition scores as indicators of values internalization neglects that values formation is a dynamic process that can be divided into stages. [53] ([53]) and [50] ([50]) assessed students’ prosocial values internalization by measuring the motivations underlying prosocial behavior (e.g., “I want to be praised for displaying good behavior”). However, these items do not provide information about an individual’s specific stage in the values internalization process, making it challenging to design targeted interventions. Second, these measurement tools were typically developed to assess the internalization of specific values. For example, [69] ([69]) created a scale specifically to measure the internalization of organizational norms, whereas [32] ([32]) developed a scale to assess students’ internalization of organizational commitment. To date, no generalized tool has been developed to measure the internalization of diverse values. Third, the reliability and validity of certain scales are insufficiently comprehensive. For instance, [69] ([69]) performed only exploratory factor analysis while developing and applying an organizational system internalization tool, without conducting confirmatory factor analysis or criterion-related validity tests. The lack of comprehensive validation limits the robustness and applicability of these measures. 

Other scholars indirectly measured individual values internalization using existing scales. [24] ([24]) and [43] ([43]) explored the relationship between parenting styles and social values internalization by employing the SVS ([56]; [66]), in which they used the dimensions of self-transcendence and conservation as indicators of social values internalization. As previously mentioned, the SVS has been widely used internationally, and research has confirmed that it broadly covers most value domains and has cross-cultural universality ([56]). Therefore, the SVS serves as a comprehensive and universally applicable instrument for assessing the degree of internalization of a broad spectrum of values. However, the use of the SVS to measure internalization has notable limitations. The SVS operationalizes values based on their perceived importance ([2]), but prioritizing a value does not necessarily reflect its genuine internalization. Individuals often fail to align consistently with the values they deem important. For example, people may endorse helping others and honesty as highly important while frequently neglecting to practice these values in daily life ([31]). However, measuring values internalization solely based on importance neglects its distinct characteristics at the motivational and behavioral levels. Therefore, while indirect tools, such as the SVS, provide useful insights into values internalization, they lack specificity. 

### 1.3. The Present Study

Therefore, it is imperative to develop psychometrically robust scales for values internalization that align with the developmental trajectories. To address the limitations mentioned above, this study proposes improvements in two key dimensions: it aims to provide tools for measuring the stages of values internalization based on a four-stage model ([75]). The four-stage model conceptualizes the process from a subjective perspective and consists of four stages: ignoring-resistance, understanding, attempt to practice, and integration. By operationalizing this process-oriented framework, we aimed to construct a scale that identifies an individual’s specific stages within the internalization continuum. Stage-specific assessments would enable targeted interventions. This tool is commonly designed to measure the process of basic human values internalization. During the development of the scale, Schwartz’s ten basic human values were adopted as measurement targets for internalizing values. In doing so, we focused on Schwartz’s theory of values for four main reasons. First, these values and their relational structure have been successfully tested in many countries, indicating that they may be universal. This universal recognition of Schwartz’s value system can serve as a basis for developing a universally applicable VIS. Second, Schwartz’s value system was developed by refining and reorganizing Rokeach’s value theory ([48]), which has been widely used internationally; research has confirmed that it broadly covers most value domains ([6]). Third, many studies on value change have adopted this theory as their theoretical framework. The development of the VIS is also based on Schwartz’s values theory, which is conducive to subsequent research on changes in values internalization levels. Fourth, Schwartz’s theory is the most cited theory in social psychology and is considered an evidence-based theory of personal values ([5]; [17]; [52]). 

In summary, this study constructed a VIS based on the four-stage model of values internalization using Schwartz’s ten basic human values. To develop the VIS, this study followed a procedure commonly employed in scale development research ([21]). First, we generated an initial item pool based on a four-stage model of values internalization and expert evaluations. Second, Sample 1 was used to verify the factorability of the data and conduct an exploratory factor analysis (EFA) to identify the underlying factor structure. Third, confirmatory factor analysis (CFA) was conducted on another independent dataset (Sample 2) to validate the factor structure and assess the model fit. Fourth, we examine the criterion-related validity of the VIS using theoretically relevant variables based on Sample 2. Finally, we assessed the internal consistency, criterion-related validity, and test–retest reliability of the VIS, with these analyses also conducted using Sample 2. Together, these steps provide robust psychometric evidence that supports the VIS as a reliable and valid instrument for assessing individuals’ stages of values internalization. 

## 2. Participants and Methods

### 2.1. Participants

Two independent participant samples were recruited for this study via Credamo (www.credamo.com), a Chinese online data collection platform similar to Qualtrics. Credamo maintains a participant pool of over 3 million individuals, covering all provinces, cities, and regions across China. Using Credamo’s random questionnaire distribution service, two samples (Samples 1 and 2) were obtained through random sampling from this pool. To minimize the risk of geographically clustered responses, a minimum distance of 10 km was set between respondents. All responses were collected anonymously to reduce the potential influence of social desirability bias on the survey results ([25]). Because the Credamo platform requires respondents to complete all items before submission, there was no missing data; thus, no missing data handling was necessary. Given that the developed scale is intended for use with the general population, participants were sampled accordingly, rather than drawn from specific or heterogeneous subgroups. According to [12] ([12]), sample size adequacy for scale development can be evaluated as follows: 100 = poor, 200 = fair, 300 = good, 500 = very good, and ≥1000 = excellent. A sample size of 500 was used for Sample 1 and Sample 2.

Sample 1 was used for item analysis and EFA, whereas Sample 2 was used for CFA, internal consistency reliability, criterion-related validity, and test–retest reliability. This study was conducted in accordance with the principles of the Declaration of Helsinki and was approved by the ethics board of Southwest University.

For Sample 1, data were initially collected from 500 participants. After excluding respondents with inconsistent responses, patterned answers, or excessively short or long response times, 474 valid responses remained, yielding a response rate of 94.8%. Among them, 151 were male (31.9%) and 323 were female (68.1%), with ages ranging from 18 to 75 years (*M* = 30.65, *SD* = 7.89). The participants’ educational levels were as follows: junior high school or below (0.4%), high school/vocational school (2.7%), associate’s degree (6.8%), bachelor’s degree (66.9%), master’s degree (21.1%), and doctoral degree or above (2.1%). 

For Sample 2, a total of 500 participants were included. After applying the same exclusion criteria as in Sample 1, 470 valid responses were obtained, yielding a response rate of 94%. Among them, 150 were male (31.9%) and 320 were female (68.1%), with ages ranging from 18 to 64 years (*M* = 30.19, *SD* = 7.69). The education levels of the sample were as follows: junior high school or below (0.2%), high school/vocational school (2.1%), associate’s degree (7.7%), bachelor’s degree (67.4%), master’s degree (20.6%), and doctoral degree or above (1.9%).

### 2.2. Research Procedure

Step 1: Development of the initial VIS. Based on the qualitative research findings of [75] ([75]), interviews identified a four-stage model of values internalization. By incorporating the definition of values internalization, a preliminary values internalization scale comprising four dimensions was developed. The item pool (containing 45 items) was reviewed by an expert panel consisting of a researcher and three psychology specialists: a PhD student specializing in values research, a PhD student with experience in scale development, and an associate professor of psychology. The experts evaluated the items based on the following criteria: (1) accurate reflection of values internalization characteristics; (2) precise representation of the four stages of values internalization; and (3) clarity and unambiguity of wording. Based on expert feedback, revisions were made to refine the scale, resulting in a 40-item preliminary version (sequentially coded as Q1-Q40). The items were distributed as follows: 10 for “ignoring-resistance stage”, 11 for “understanding stage”, 8 for “attempt to practice stage”, and 11 for “integration stage”. A 5-point Likert scale was used for responses.

Step 2: Finalization of guiding statements for VIS. Using Schwartz’s ten basic values (self-direction, stimulation, hedonism, achievement, power, security, conformity, tradition, benevolence, and universalism) as the targeted values in the questionnaire development, each value was represented by a specific instructional statement. The instructions used a descriptive sentence to introduce the values, as exemplified in Step 3. Before data collection, three psychology graduate students (two PhD students and one master’s student) reviewed the 10 instructional statements of the Initial VIS to ensure their accurate representation of intended meanings.

Step 3: Testing with Sample 1. A 40-item preliminary scale was administered to Sample 1. Sample 1 included 500 participants who were randomly assigned to 10 groups corresponding to 10 values. Each group completed the scale under the corresponding value instructional statement. All participants responded to the same 40 items, which were presented in a random order across all values. 

Taking self-direction as an example, the test process is as follows: First, participants were presented with the instruction of self-direction: “Thank you for participating in this survey. Please consider the following statement: ‘I will think and act independently, including choosing, creating, and exploring, in the form of freedom, independence, and self-determination.’ Based on how well this value applies to you, indicate your level of agreement with each of the following items by selecting the corresponding number from the options: 1 = Strongly Disagree, 2 = Disagree, 3 = Uncertain, 4 = Agree, and 5 = Strongly Agree. Note that everyone holds unique values; therefore, there are no correct or incorrect answers. We encourage you to respond as accurately and truthfully as possible.” Participants were then asked to answer 40 items, such as “I resist this value” (see Table 2 for examples), by selecting the corresponding number. Finally, the participants provided demographic information, including age and sex. Upon completion of the questionnaire, participants received a designated reward. 

Step 4: Testing with Sample 2. Based on the results of the item analysis and EFA, the revised version of the VIS, along with the criterion-related validity questionnaire, was administered to Sample 2. Schwartz’s ten basic values and the corresponding guiding statements established in Step 2 were used. During the recruitment of Sample 2, the participants were informed that the study would include a follow-up survey. All 500 participants consented to participate in the subsequent investigation. A total of 500 participants were randomly assigned to ten groups, each corresponding to one of the ten target values. Under the guidance of the assigned value group, each participant first completed the corresponding version of the VIS. Subsequently, the participants completed a criterion-related validity questionnaire and provided their demographic information. A small monetary reward was offered upon the completion of the questionnaire. 

Step 5: Five weeks after completing the survey for Sample 2, a follow-up assessment was conducted to evaluate the test–retest reliability of the VIS. Using the Credamo platform, a follow-up questionnaire—identical to the VIS administered in Step 4—was distributed to participants in Sample 2. Owing to time constraints and participant attrition, 194 individuals participated in the retest. After excluding responses that exhibited inconsistencies, patterns, or abnormally short or long completion times, 184 valid responses remained. The demographic characteristics of the participants were as follows: 53 were male (28.8%) and 131 were female (71.2%), with ages ranging from 19 to 55 years (*M* = 29.77, *SD* = 6.682). The educational levels of the participants were as follows: junior high school or below (0.4%), high school/vocational school (2.2%), associate’s degree (9.2%), bachelor’s degree (25.5%), master’s degree (25.5%), and doctoral degree or above (0.5%).

### 2.3. Criterion Measures

Protected Values Scale. The Protected Values Measurement Questionnaire, developed by [68] ([68]), was used in this study. This scale consists of five items scored on a 7-point Likert scale, where 1 represents “strongly disagree” and 7 represents “strongly agree.” The total score across the five items represents the degree of protection associated with the given values. In the present study, the Cronbach’s α coefficient for this scale was 0.694.

Attitude Strength Scale. This study adopts the Attitude Strength Scale used by [46] ([46]). The scale consists of six items rated on a 9-point continuum from −4 to 4, evaluating four dimensions: attitude certainty, attitude importance, self-involvement with the attitude, and knowledge related to attitudes. These six items have been widely used in research on attitude strength. In this study, the Cronbach’s α coefficient for the scale was 0.758. 

## 3. Results

Data analysis was conducted using SPSS (version 25.0) for item analysis, exploratory factor analysis, reliability, and validity testing, while Mplus 8.3 was utilized for confirmatory factor analysis.

### 3.1. Item Analysis

Homogeneity testing was used as an indicator for item analysis. This test examined the relationship between each item and its corresponding subscale total scores to assess the degree of homogeneity. Items with correlation coefficients lower than 0.4 were removed ([23]). Based on the analysis results, four items were deleted: “To avoid punishment, I began to pay attention to this value” (*r* = 0.154), “Through media publicity, I gradually became aware of this value” (*r* = 0.007), “I know that practicing this value is meaningful” (*r* = 0.301), and “Practicing this value is painful” (*r* = 0.033). The remaining items were significantly correlated with their respective subscale scores, with correlation coefficients ranging from 0.446 to 0.832.

### 3.2. Exploratory Factor Analysis

An EFA was conducted on the remaining 36 items to assess the factor structure and reduce the overall number of items. Bartlett’s test of sphericity (χ^2^ = 12,346.451, *df* = 780, *p* < 0.001) and the Kaiser–Meyer Olkin (KMO) measure of sampling adequacy (KMO = 0.966) indicated that the items shared potential underlying factors and were suitable for factor analysis ([67]). The principal axis factoring method and varimax rotation were used. The number of factors was determined based on an eigenvalue greater than 1 ([30]) and a scree plot test ([47]). Items were sequentially deleted based on the following criteria, with factor analysis re-run after each deletion: (1) items with absolute factor loadings below 0.4 ([10]); (2) items with severe cross-loadings ([10]); (3) items loaded onto a factor inconsistent with their conceptual dimensions ([45]); and (4) ensuring that each factor retained at least three items ([10]). After removing 11 items, a final scale comprising 25 items across four dimensions was established. The total variance explained by the four factors was 56.855%, and the communalities of the 25 items ranged from 0.411 to 0.666. Table 2 presents the results of EFA.

Based on the meanings of the items within each factor and [75] ([75]) qualitative research findings, the four dimensions were labeled as: ignoring-resistance, understanding, attempt to practice, and integration stages. Ignoring-resistance stage (seven items): During this stage, the individual has not internalized the value. In addition, the individual is unaware of the importance of the value, does not endorse it, or actively rejects it, potentially adopting an opposing stance. Understanding Stage (six items): after re-evaluating the value, individuals overcome their resistance or recognize its importance and begin to understand it. Attempt to practice stage (five items): following a shift in value cognition, the individual attempts to apply these values under the influence of external motivation while progressively deepening their understanding of such values. Integration Stage (seven items): representing advanced internalization, where individuals autonomously internalize the value, abandon prior conflicting beliefs, and consistently guide their behavior through it. 

### 3.3. Confirmatory Factor Analysis

To assess structural validity, CFA was conducted using the maximum likelihood estimation (ML) method ([8]), following these steps: (1) constructing a single-factor model (M1) in which all items were loaded onto a single undifferentiated factor, and (2) constructing a four-factor model (M2) based on the EFA results. The results indicated that the fit indices for the single-factor model (M1) were χ^2^/*df* = 9.256, *p* < 0.001, RMSEA = 0.133, 90%CI [0.128,0.137], CFI = 0.637, TLI = 0.604, and SRMR = 0.121. For the four-factor model (M2), the fit indices were: χ^2^/*df* = 2.90, *p* < 0.001, RMSEA = 0.064, 90%CI [0.059,0.069], CFI = 0.918, TLI = 0.909, SRMR = 0.058. According to established fit indices, CFI and TLI > 0.90 ([4]), RMSEA and SRMR ≤ 0.08 ([9]), and χ^2^/*df* between 1 and 3 ([55]), the four-factor model (M2) demonstrated a significantly better fit than the single-factor model (M1), with all indices falling within the acceptable range. The structural representation of the four-factor model of the Values internalization Scale is shown in Figure 2, with factor loadings ranging from 0.623 to 0.834. 

### 3.4. Internal Consistency Reliability

The internal consistency reliability of the four dimensions of the VIS was assessed using Cronbach’s α coefficients with Sample 2 (*N* = 470). The results showed that Cronbach’s α coefficients for the four dimensions—ignoring-resistance, understanding, attempt to practice, and integration stages—were 0.898, 0.862, 0.868, and 0.876, respectively. According to [22] ([22]), a Cronbach’s α coefficient between 0.70 and 0.80 is considered good, while a coefficient between 0.80 and 0.90 is considered very good. Thus, each dimension of the scale demonstrated excellent internal consistency reliability.

### 3.5. Criterion-Related Validity

Simultaneously, a criterion-related validity analysis was conducted using Sample 2 (*N* = 470). The Pearson correlation coefficients between the scores for each dimension of the VIS and the two criterion variables (the Protected Values Scale and Attitude Strength Scale) are presented in Table 3. The four dimensions of the VIS were significantly correlated with both Protected Values and Attitude Strength scores (*p* < 0.05), indicating that the VIS demonstrated good criterion-related validity. Specifically, ignoring-resistance and understanding stages exhibited negative correlations with the intensity of protected values and attitude strength (*p* < 0.01). In contrast, the attempt to practice and integration stages showed positive correlations with the intensity of protected values and attitude strength (*p* < 0.05). Furthermore, as the values progressed to higher stages of internalization, the correlation coefficient between the values internalization stage and the criterion questionnaire increased (see Figure 3). This suggests that as the internalization of values increases, the degree to which values become protected becomes deeper, and the attitude toward values becomes more stable. 

### 3.6. Test–Retest Reliability 

A subset of 184 participants from Sample 2 (*N* = 184) was randomly selected for test–retest reliability assessment after a five-week interval. The test–retest reliability coefficients for the four dimensions of the scale were 0.602, 0.622, 0.687, and 0.648, respectively, all of which exceeded the threshold of 0.60. According to [15] ([15]), correlation coefficients between 0.4 and 0.7 indicate a moderate correlation, while those between 0.7 and 0.9 indicate a strong correlation. Based on this criterion, the test–retest reliability of all four dimensions of the scale met acceptable psychometric standards. Additionally, the test–retest reliability coefficients of the four dimensions were consistent with similar studies, such as the studies by [36] ([36]), who reported an average test–retest reliability coefficient of 0.65 for values in the SVS, and [3] ([3]), who reported an average test–retest reliability of 0.64 for Schwartz’s ten basic values. These findings indicated that the four dimensions of the scale exhibited good temporal stability.

## 4. Discussion

Building on qualitative research on values internalization, this study explores the structural composition of the values internalization process, following psychometric principles and procedures. First, based on the representative items identified in the qualitative research of [75] ([75]) and the theoretical foundation of the four-stage model of values internalization ([75]), a scale structure and preliminary items were established. Relevant experts were invited to review and refine the items. Subsequently, a series of psychometric evaluations, including item analysis, EFA, and CFA, were conducted. The results indicated that the four-factor structural model demonstrated a good fit and that the questionnaire exhibited satisfactory internal consistency reliability, criterion-related validity, and test–retest reliability. These findings confirm that the VIS developed in this study is valid and reliable. Ultimately, a 25-item scale encompassing four dimensions was established (see Appendix A: Values internalization scale).

First, the results of the EFA and CFA indicate that the values internalization process follows a four-factor structure: ignoring-resistance stage, understanding stage, attempt to practice stage, and integration stage. Furthermore, the four dimensions of the VIS developed in this study are substantiated by previous theoretical and empirical research. Specifically, the ignoring-resistance stage refers to a phase in which individuals have not yet begun internalizing values and remain in a negative state of either neglecting or resisting them. While no prior studies have directly investigated this phenomenon, [34] ([34]) cognitive development theory suggests that individuals at different developmental stages exhibit varying attitudes toward social norms and values. Some individuals may ignore social norms or resist particular values, especially when facing social pressures or cultural conflict ([27]; [70]). The understanding stage describes individuals who begin to reassess, accept, and comprehend their values ([73]; [29]). Previous studies have also demonstrated that cognitive reappraisal and acceptance in daily life are associated with value development ([35]). The attempt to practice stage is when individuals try to act on values they have not yet fully internalized. Similar stages have been found in existing theories of values internalization. For instance, in the three-stage theory of [33] ([33]), “compliance” and “identification” refer to individuals who follow values under external motivational influences. Finally, the integration stage refers to individuals who actively practice, identify with, and stably hold values over time. This aligns with the internalization stage in [33] ([33]) theory and the integrated regulation stage in self-determination theory, both of which describe individuals as fully incorporating values into their personal value systems, marking the highest level of values internalization. 

Second, the four dimensions of the VIS were significantly correlated with the scores of both protected values and attitude strength. The correlation coefficients between the values internalization stages and attitude strength increased as the internalization stage progressed, suggesting that higher stages of internalization are associated with more firmly held attitudes. Scholars have proposed that a direct connection between values and attitudes strengthens attitude intensity ([64]). Values are often described as guiding principles in one’s life ([60]) and have been empirically demonstrated to shape both attitudes and behaviors ([28]). Additionally, the correlation coefficient between the values of the internalization stage and protected values increased with the progression of the stages, suggesting that as individuals progress through internalization, their values become more aligned with protected values. [76] ([76]) proposed that value becomes most important to an individual only after self-internalization, which reduces cognitive weighing and makes the value more resistant to change. Therefore, in the criterion-related validity test of this study, the relationships between values internalization stages and criterion variables were consistent with prior research, demonstrating that the VIS demonstrates satisfactory criterion-related validity. 

Third, the test–retest reliability coefficients of the VIS ranged from 0.60 to 0.69, indicating moderate temporal stability. This stability can be attributed to several factors. First, the retest sample, while demographically similar to the full sample, consisted predominantly of young adults (51.5% aged between 19 and 30 years), whose values and internalization stages may be more fluid and context-sensitive ([1]; [63]). Second, the process of values internalization is conceptualized as a dynamic developmental sequence. Although these stages possess trait-like characteristics, they are not immutable personality traits; rather, they reflect evolving engagements with values that can shift with new experiences, environments, or self-reflections ([20]; [58]). Thus, moderate stability aligns with the theoretical premise that individuals may progress or regress in stages over time, especially in earlier phases such as understanding or attempt to practice. Therefore, these coefficients may reflect developmental characteristics.

This study makes both theoretical and practical contributions. Theoretically, it advances research on the developmental nature of values internalization by offering a process-sensitive measurement tool. The VIS has demonstrated excellent reliability and validity. Existing scales used by scholars to measure values internalization typically reflect static assessments of individual values but fail to consider the dynamic developmental state of internalized values. For instance, the SVS ([56]) serves as an indicator of the level of internalization of individual values. However, it does not reveal an individual’s position within the values internalization process. Historically, internalization scales have primarily been developed to assess specific domains or particular values. By contrast, the VIS developed in this study provides a broadly applicable measure of values internalization across fundamental human values. This contributes to comparative research on differences in values internalization across individuals or groups for various values. Practically, the VIS enables a more nuanced assessment of individuals’ positions within the internalization process, which can inform longitudinal studies and intervention designs. The VIS can guide tailored educational or organizational programs to promote value adoption at different stages. For example, those in the “understanding” stage may benefit more from reflective exercises or persuasion manipulation, while those in the “attempt to practice” stage may require support for behavioral consistency.

Nevertheless, several limitations must be acknowledged, which also provide directions for future research: (a) The VIS developed in the present study provides a tool for facilitating more sophisticated and fine-grained investigations into the dynamics of values internalization. Recognizing that internalization is a continuous and non-linear process, where individuals may skip stages, regress, or repeatedly transition between phases depending on personal, situational, and cultural factors ([20]; [58]), the VIS enables the empirical examination of such complex trajectories and individual differences. Although the four-stage model proposed by [75] ([75]) offers a theoretical foundation, ongoing research is required to further validate the model. Thus, the VIS is positioned to support future efforts to capture the nuanced and evolving nature of values internalization. (b) VIS was developed and validated using Chinese samples. Given the cultural sensitivity of values ([59]) and the influence of sociocultural norms and personal experiences on individual value systems ([37]), generalizability may be limited. For example, Western cultures often emphasize self-oriented values, such as personal achievement, independence, and freedom, whereas Eastern cultures typically prioritize other-oriented values, such as social harmony, interpersonal cooperation, and collective well-being ([56]). Future studies should explore potential cultural variations in the stages of values internalization and examine the cross-cultural applicability of the VIS. (c) Developmental variations in values internalization across life stages were not examined. Prior research suggests that adolescence is a key period for value formation ([16]), whereas values continue to develop in adulthood, albeit slower than in previous developmental stages ([65]). It is plausible that individuals of different age groups progress through internalization stages at varying rates or in different orders. Future studies should investigate age-related patterns to enhance the developmental validity of the VIS. (d) In the current sample, there were more female than male participants. Although values internalization is not assumed to be sex-specific, gender differences in value orientation and self-construal have been documented ([62]). Future studies should aim to achieve a more balanced sex representation to ensure generalizability. (e) Values differ substantially in the level of importance individuals assign to them ([56]), and these differences may influence the internalization process. Specifically, values perceived as more important may progress more rapidly or smoothly through the internalization stages than less important values. Future studies could systematically compare how the internalization of values with differing levels of subjective importance varies across stages. (f) Beyond anonymous responses, future research could incorporate direct measures of social desirability, such as the Marlowe–Crowne Social Desirability Scale ([13]), to statistically control for this bias. (g) The test–retest reliability was only moderate, and the follow-up survey experienced a high attrition rate, with the remaining participants largely concentrated in younger age groups. Future studies should conduct longitudinal assessments to address these limitations. In conclusion, this study developed and validated a novel and psychometrically sound instrument to assess the process of values internalization. While preliminary findings are promising, further theoretical refinement, cross-cultural validation, and longitudinal tracking are necessary to fully realize the potential of the scale.

## Figures and Tables

**Figure 1 behavsci-15-00660-f001:**
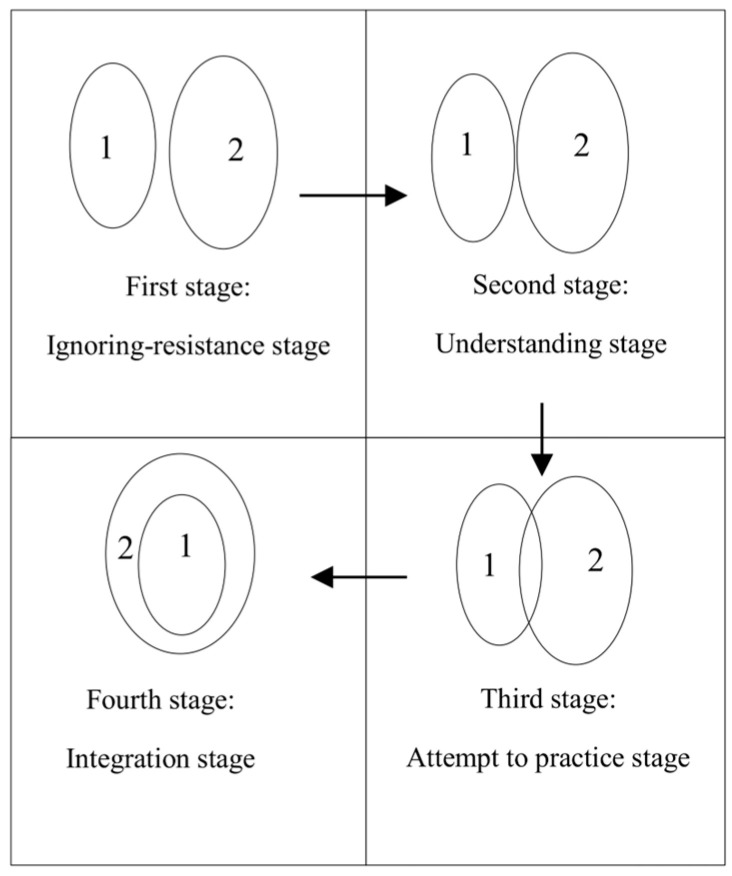
Four-stage model of values internalization from [75] ([75]). Notes: “1” represents external values; “2” represents the individual’s internal values system.

**Figure 2 behavsci-15-00660-f002:**
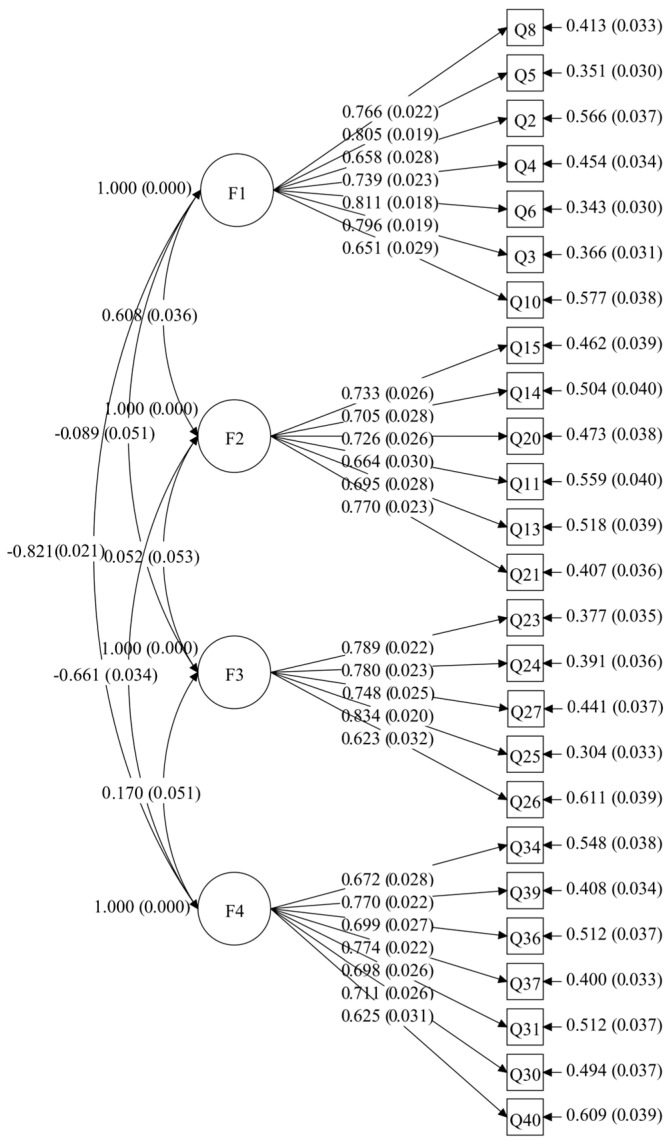
The four-factor structure of the values internalization scale. Note: F1 = ignoring-resistance stage; F2 = understanding stage; F3 = attempt to practice stage; F4 = integration stage.

**Figure 3 behavsci-15-00660-f003:**
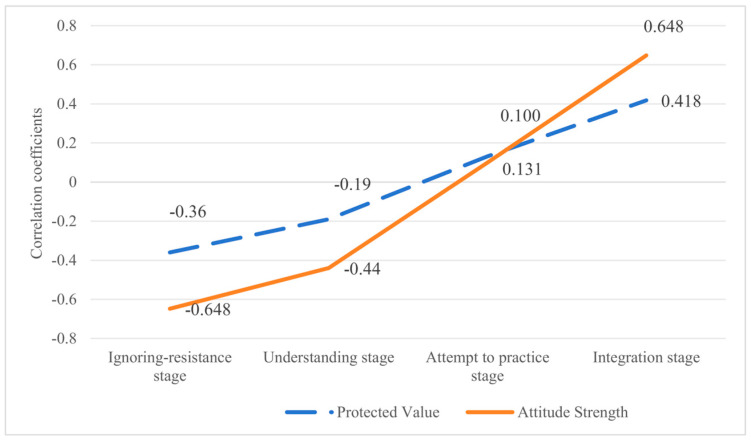
Correlation coefficients between the four stages of values internalization and criterion measures.

**Table 1 behavsci-15-00660-t001:** Characteristics of the four stages of values internalization from [75] ([75]).

	Stages	Ignoring-Resistance Stage	Understanding Stage	Attempt to Practice Stage	Integration Stage
Characteristics					
Cognition		Rejecting values information	Initial understanding of values (superficial understanding)	/	Values endorsement; Adhering to values
Behavior		Behavior inconsistent with values	Accept values but have difficulty adhering to them	Difficulty in practicing values	Efforts to practice values; Behavior guided by values
Motivation		/	/	External motivation affects behavior	Enhance initiative to align with values; Internal motivation
Emotion		Negative emotions toward values	/	Fear of practicing values	Enjoyment of a value-aligned life
Self-relevance		Low relevance of values to life; Low importance of values	Low priority of values	/	Values are related to self; High importance and priority

Notes: “/” represents the lack of clearly defined characteristics in the corresponding dimension at that stage.

**Table 2 behavsci-15-00660-t002:** Factor loadings of values internalization scale items (*N* = 474).

Item Number	Items	M (*SD*)	Ignoring-Resistance Stage	Understanding Stage	Attempt to Practice Stage	Integration Stage	Communality
1	Q8 I reject this value.	1.75 (1.002)	0.752				0.666
2	Q5 I will not follow this value even if there is a reward.	1.78 (0.958)	0.745				0.629
3	Q2 I don’t understand this value.	1.77 (0.842)	0.669				0.509
4	Q4 My actions are contrary to this value.	1.80 (0.981)	0.710				0.623
5	Q6 Even if ordered by others, I am unwilling to act according to this value.	1.91 (1.042)	0.704				0.623
6	Q3 I would not consider practicing this value.	1.90 (1.024)	0.702				0.635
7	Q10 This value is remote from my life.	1.83 (0.958)	0.612				0.526
8	Q15 This value is meaningful, but I have not yet applied it to my life.	2.11 (1.065)		0.775			0.612
9	Q14 Although I value this belief, I have not practiced it.	2.03 (1.011)		0.701			0.516
10	Q20 This value is important, but I do not adjust my behavior based on it.	1.97 (0.995)		0.635			0.521
11	Q21 This value is important, yet I do not adhere to it as a personal standard.	2.05 (1.039)		0.606			0.530
12	Q13 I often forget that my behavior should align with this value.	2.01 (0.976)		0.563			0.433
13	Q11 I don’t have a clear understanding of this value.	2.11 (0.956)		0.542			0.439
14	Q23 I practice this value to maintain relationships with others.	3.16 (1.266)			0.787		0.636
15	Q27 I try to put this value into action to gain profit.	2.96 (1.309)			0.773		0.608
16	Q25 I act on this value to avoid punishment.	2.58 (1.224)			0.770		0.608
17	Q24 I start practicing this value to gain others’ approval or appreciation.	3.36 (1.267)			0.745		0.633
18	Q26 I attempt to act on this value due to external supervision.	3.20 (1.200)			0.654		0.453
19	Q40 This value is more important than anything else.	3.79 (1.074)				0.627	0.610
20	Q39 This value is the foundation for other things.	3.89 (1.039)				0.638	0.624
21	Q34 When I violate this value, I engage in compensatory behaviors.	3.83 (1.113)				0.630	0.646
22	Q36 I take the initiative to spread this value to others.	3.89 (1.112)				0.607	0.613
23	Q37 I have negative feelings when I violate this value.	3.47 (1.156)				0.491	0.411
24	Q31 I will not change this value when facing external temptations.	3.76 (1.065)				0.521	0.511
25	Q30 I have a deep understanding of this value.	3.96 (0.984)				0.538	0.518
	Rotated Eigenvalues		4.810	3.225	3.193	2.986	
	Variance Explained (%)		19.240%	12.898%	12.771%	11.946%	

**Table 3 behavsci-15-00660-t003:** Correlation analysis between values internalization dimensions and criterion variables (*N* = 470).

	1	2	3	4	5	6
1 Ignoring-resistance stage	1.00					
2 Understanding stage	0.541 **	1.00				
3 Attempt to practice stage	−0.087	0.053	1.00			
4 Integration stage	−0.647 **	−0.574 **	0.132 **	1.00		
5 Protected Values	−0.360 **	−0.190 **	0.131 **	0.418 **	1.00	
6 Attitude Strength	−0.648 **	−0.440 **	0.100 *	0.648 **	0.425 **	1.00

Note: ** *p* < 0.01; * *p* < 0.05 (two-tailed test).

## Data Availability

The data used for this analysis will become available through the first author at any time from now upon reasonable request.

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
