# Peer review of "Development and Validation of the Values Internalization Scale"

_behavsci, 2025, doi:10.3390/bs15050660_

Round 1
Reviewer 1 Report
Comments and Suggestions for Authors
1 Introduction
What are the purpose and objectives of the paper? This needs to be articulated much earlier for the reader. The paper starts very open and broadly, with too many general information in the beginning. What is your message, where is the focus, research gap? Your introduction is rather a literature review; thus the actual introduction is actually missing.
Throughout the paper look out for inconsistency in usage of terminology: e.g. "value internalization" and "values internalization". Further, the term internalization is introduced with various slightly different definitions from several researchers (e.g., Deci & Ryan, Grusec & Goodnow), but there is no clear, unified working definition adopted that you follow throughout the paper.
Vygotsky’s sociocultural theory, Kelman’s model, and SDT are all introduced as stage-based models. However, the link between these models and value internalization specifically is not similarly strong. In the introduction you mention to measure the internalization of “diverse values,” but it is not clear whether that includes moral values, professional norms, religious beliefs, or more. Internalization processes are also expected to change across life stages (e.g., adolescence vs. adulthood), which is not addressed here.
Wu et al.'s four-stage model is presented but lacks a critical stance. What is also missing is a clear discussion of empirical support, methodological limitations, or comparison to existing theories. Also, alternative ways are not take into consideration.
You should justify better why you base the new tool on Schwartz’s 10 values; why do you choose exactly these values for a process model? What is also missing are potential challenges in developing or validating a new psychometric tool; also practical and theoretical values should be more pronounced.
What is missing in your introduction is mentioning methods/measurement design; also structure of the whole paper is not mentioned nor outlined.
Figuer 1 is somewhat blurred. Also the set-up of table 1 is unusual and not nice to read.
2 Participants and Methods
Overall, this section is very brief and needs more information. It is mentioned that participants were recruited via Credamo, but little information about the programme, so how diverse is the participant pool, what are possible limitations of online panel? How were participants selected, by random sampling or stratified by age? Why are there many more women taking part? What implications does it give us?
You write that “each of Schwartz’s 10 values was randomly assigned to 50 participants”, but it does not become clear whether the same set of items was used with different values or whether items were value-specific. Was there any pre-testing of the questionnaire? It is also unclear how the four-stage model was tested structurally: based on stage categories or were stage-specific models compared in CFA? How did you deal with social desirability bias in your study? How did you deal with missing or incomplete values?
3 Results
The study removes items with subscale correlations <0.40 but you do not really explain why you chose 0.40. It could be that different thresholds are more suitable of course depending on sample size, construct density, or item purpose. PCA was used with varimax rotation, which is more of a data reduction method than a actual latent factor extraction method. Further, you only test two models, but no alternative structures such as 2- or 3-factor.
Does the retest sample reflects the demographics of the full sample? What is more, all test-retest coefficients are between 0.60–0.69, which is of course moderate, but not best. So you should mention why stability is only moderate and what it means for interpreting stages as trait-like.
4 Discussion
The discussion is a lot of repetition from the results chapter without critically engaging with the results, meaning, or implications. Further, stages are discussed as clearly distinct, but no acknowledgement that in practice, people may connect stages, revert, or progress non-linearly. Also, four-stage model is justified without considering whether the data might also support a dimensional or hierarchical model. The results show variance in scores, but the discussion treats the process as universal rather than shaped by personal factors (e.g., age, education). It is not questioned if qualitative findings can be transferred to the psychometric instrument, as supposed by Wu et al. (2025). What is also lacking is discussing more about theoretical or practical implications, such as explaining better certain interventions, or future research. As such at least it should be proposed to test the model longitudinally. You briefly mention that the study was conducted in China and talk about limitations, however, the rest of the discussion assumes the model is universally applicable.
Comments on the Quality of English Language
can be improved
Reviewer 2 Report
Comments and Suggestions for Authors
Dear Authors,
The submitted article has been carefully prepared and appears to be appropriate in terms of its background, objectives, and methodology. However, a few minor suggestions and comments are offered for your consideration.
Abstract
In the abstract, it would be useful to explain the context of the study and possibly indicate the area of application, as this is not clear from the initial introduction. Please also consider whether it would be appropriate to specify the area of socialization mentioned. Please indicate whether the newly developed scale is suitable for the adult population, pupils, students.
Introduction
Please consider whether it would be more appropriate to use the past tense in line 145.
Participants and Methods
Please include in section 2.1 the definition of the theoretical population from which the sample was drawn and the specific sampling technique used (convenience, random, quota, etc.). Given the diversity of readers, it would also be useful to briefly explain the (minor) role of sample representativeness in this type of study.
Line 193 mentions the "item pool"; please add information about the total number of statements tested.
For the information on lines 196-202, please clarify the order in which the items were presented to the participants. Was randomization used in this case?
Is there any information on the adherence and perceived importance of the ten core values reported on lines 203-204? If so, please provide it.
Please clarify the relationship between the guiding words on line 207 and the basic values.
Please clarify whether the guiding words on lines 208-210 refer to Sample 1.
Please check the translation of the task on lines 213-217. The current form of the answer options does not seem to correspond to the wording of the question.
Please consider adding a new table with basic descriptive statistics for each item (especially means, standard deviation, etc.).
Discussion
It would be useful to add and discuss further limitations of the study conducted. In particular, the sensitivity of the developed scale with respect to specific values is important: How might the results be affected by differences in the importance of specific values from the perspective of the respondents? In addition, it would be useful to mention the issue of social desirability when responding to individual items.
Please consider including information about the practical applicability of the scale.
Thank you for your consideration of these comments, and I wish you well in your future work.
Sincerely,
Round 2
Reviewer 1 Report
Comments and Suggestions for Authors
Thank you for the revision and answer letter.